# Genome-Inspired Chemical Exploration of Marine Fungus *Aspergillus fumigatus* MF071

**DOI:** 10.3390/md18070352

**Published:** 2020-07-06

**Authors:** Jianying Han, Miaomiao Liu, Ian D. Jenkins, Xueting Liu, Lixin Zhang, Ronald J. Quinn, Yunjiang Feng

**Affiliations:** 1Griffith Institute for Drug Discovery, Griffith University, Brisbane, QLD 4111, Australia; jianying.han@griffithuni.edu.au (J.H.); miaomiao.liu@griffith.edu.au (M.L.); i.jenkins@griffith.edu.au (I.D.J.); 2Key Laboratory of Pathogenic Microbiology and Immunology, Institute of Microbiology, Chinese Academy of Sciences, Beijing 100101, China; lzhang03@gmail.com; 3State Key Laboratory of Bioreactor Engineering, East China University of Science and Technology, Shanghai 200237, China; liuxt2010@126.com

**Keywords:** *Aspergillus fumigatus*, genome mining, chemical diversity, antimicrobial activity, biosynthetic gene cluster, prenyltransferase

## Abstract

The marine-derived fungus *Aspergillus*
*fumigatus* MF071, isolated from sediment collected from the Bohai Sea, China, yielded two new compounds 19*S*,20-epoxy-18-oxotryprostatin A (**1**) and 20-hydroxy-18-oxotryprostatin A (**2**), in addition to 28 known compounds (**3**–**30**). The chemical structures were established on the basis of 1D, 2D NMR and HRESIMS spectroscopic data. This is the first report on NMR data of monomethylsulochrin-4-sulphate (**4**) and pseurotin H (**10**) as naturally occurring compounds. Compounds **15**, **16**, **20**, **23**, and **30** displayed weak antibacterial activity (minimum inhibitory concentration: 100 μg/mL). Compounds **18** and **19** exhibited strong activity against *S. aureus* (minimum inhibitory concentration: 6.25 and 3.13 μg/mL, respectively) and *E. coli* (minimum inhibitory concentration: 6.25 and 3.13 μg/mL, respectively). A genomic data analysis revealed the putative biosynthetic gene clusters *ftm* for fumitremorgins, *pso* for pseurotins, *fga* for fumigaclavines, and *hel* for helvolinic acid. These putative biosynthetic gene clusters fundamentally underpinned the enzymatic and mechanistic function study for the biosynthesis of these compounds. The current study reported two new compounds and biosynthetic gene clusters of fumitremorgins, pseurotins, fumigaclavines and helvolinic acid from *Aspergillus*
*fumigatus* MF071.

## 1. Introduction

It has been demonstrated that marine-derived microbes have become one of the most important sources of pharmacologically active metabolites [1,2,3,4,5]. Under extreme marine conditions, such as high salinity, high pressure, low temperature, and extreme pressures, microbes have evolved unique physiological and chemical capabilities to survive and proliferate [6]. Marine natural products (MNPs) display a wide range of structural diversity and remarkable pharmaceutically relevant bioactivities, including antibacterial, antiviral, anticancer, and anti-inflammatory properties [7,8,9]. Fungi derived MNPs are the largest category among all marine sources (bacteria, cyanobacteria, algae, sponges, invertebrate, and mangroves), with the average number of compounds in 2018 increased by 85% compared with the previous three years (2015–2017) [8].

The increasing rediscovery rate of known compounds through high-throughput screening (HTS) has led to a decline in natural product research, whereas both hospital and community-associated infectious pathogens with antimicrobial resistance (AMR) are spreading rapidly [10]. With technological advances in microbial genome sequencing and the development of bioinformatics tools [11], genome mining approaches have revealed a hidden reservoir of untapped biosynthetic gene clusters (BGCs) [12,13]. Recent studies of large-scale and meta-genome mining have highlighted unprecedented capabilities for biosynthesis of diverse novel classes of active natural products from microbes [14,15,16].

In the course of searching for bioactive metabolites produced from marine-derived fungi, a marine microbe library and a microbial crude extracts library were constructed and screened for various biological activities [17,18,19]. One strain, *Aspergillus fumigatus* MF071, was highlighted for a chemical constituent study based on HPLC chemical profiling and NMR fingerprinting analyses. Meanwhile, the genome of MF071 was sequenced for the metabolic potential analysis. The genome-inspired chemical constituents study and biological activities investigation of an extract of MF071 led to the isolation of 30 compounds, including the two new compounds 19*S*,20-epoxy-18-oxotryprostatin A (**1**) and 20-hydroxy-18-oxotryprostatin A (**2**), along with seven active compounds. The NMR data of two compounds, monomethylsulochrin-4-sulphate (**4**) and pseurotin H (**10**), are also reported here for the first time. The isolated compounds include the structural classes of indole alkaloids, polyketide and non-ribosomal peptide hybrids, terpenoids, and polyketides. An analysis of genome sequences revealed BGCs and the biosynthetic pathway of these compounds. The activities of isolated compounds against *Mycobacterium smegmatis*, *Staphylococcus aureus*, *Escherichia coli*, and *Pseudomonas aeruginosa* were also evaluated.

## 2. Results

### 2.1. Characterization and Identification of Strain MF071

The identification of MF071 was performed based on the morphology and phylogenetic analysis. After incubation at 28 °C for ten days, strain MF071 formed colonies on a PDA plate with characteristic hyphal structures (Figure 1a) [20]. The ITS gene region of ribosomal DNA of the strain was PCR-amplified and sequenced. The phylogenetic tree (Figure 1b) constructed from the ITS gene sequence indicated that MF071 belonged to the genus of *Aspergillus* with the highest similarity to *A. fumigatus* (99.82%, accession number: EF669985). The nucleotide sequence of the ITS gene has been deposited in GenBank (accession no. MN700176). The MF071 strain has been deposited at Dr. Zhang’s Laboratory, East China University of Science and Technology.

### 2.2. Structure Elucidation of the Isolated Compounds

The chemical profiling of the MF071 extract obtained from rice medium fermentation was analysed using HPLC and ^1^H NMR (Figure 2). The ^1^H NMR displaying a wide range of signals indicated the chemical diversity of the extract. The MF071 extract was fractionated and purified, as detailed in Materials and Methods. Thirty compounds were identified, including two new compounds 19*S*,20-epoxy-18-oxotryprostatin A (**1**) and 20-hydroxy-18-oxotryprostatin A (**2**). The NMR data of the compounds monomethylsulochrin-4-sulphate (**4**), and pseurotin H (**10**) (Figure 3) are also reported.

Compound **1** was isolated as a yellow powder. Its HRESIMS revealed a molecular ion peak of *m/z* 434.1689 for [M + Na]^+^ (calculated for C_22_H_25_N_3_O_5_ Na, 434.1686), indicating a molecular formula of C_22_H_25_N_3_O_5_ (Appendix A). The UV spectrum of **1** showed maximal absorbance at 260 nm and 350 nm in MeOH. The ^1^H, ^13^C and 2D NMR (Table 1 and Appendix A) revealed the presence of three aromatic methine carbons (*δ*_C_/*δ*_H_ 123.0/7.68, d, *J* = 8.9 Hz; 94.1/6.89, d, *J* = 2.3 Hz; 112.5/6.77, dd, *J* = 8.9 and 2.3 Hz); one methoxyl group at *δ*_H_ 3.81 (*δ*_C_ 55.6); two methyl groups at *δ*_H_ 1.48 (*δ*_C_ 24.7, C-21) and *δ*_H_ 1.17 (*δ*_C_ 18.6, C-22); four methylene groups (*δ*_C_/*δ*_H_ 26.0/3.62 and 3.32, m; 45.5/3.39 and 3.28, m; 28.1/2.07 and 1.72, m; 22.6/1.75, m); three methine groups at *δ*_H_ 4.42 (*δ*_C_ 56.1, C-9), *δ*_H_ 4.36 (*δ*_C_ 63.9, C-19), and *δ*_H_ 4.16 (*δ*_C_ 58.9, C-12); and three carbonyl groups (*δ*_C_ 187.2, C-18; *δ*_C_ 167.2, C-11; *δ*_C_ 165.9, C-17).These NMR data suggested compound **1** as a tryprostatin derivative [21], including the moieties of the indole, diketopiperazine, and prenyl part. In comparison with the NMR data of 18-oxotryprostatin A (**3**) [22], the HMBC correlations from H-21 (*δ*_H_ 1.48) and H-22 (*δ*_H_ 1.17) to C-19 (*δ*_C_ 63.9) and C-20 (*δ*_C_ 61.8), and a degree of unsaturation suggested the presence of an epoxy moiety between C-19 and C-20. Thus, the planar structure of **1** was assigned as depicted (Table 1 and Figure 3). The relative configuration of compound **1** was determined by its biogenetic origin and DFT NMR calculation. The genome sequences of MF071 revealed that compound **1** had the same biosynthetic pathway as 18-oxotryprostatin A (**3**) (Figure 4B). Considering the same biogenetic origin, the relative configurations of positions C-9 and C-12 in compound **1** were proposed to be the same as those in 18-oxotryprostatin A (**3**). The ^13^C NMR chemical shifts of two possible isomers with 19*S*,20-epoxy and 19*R*,20-epoxy configurations were calculated using DFT. A DP4 probability analysis of calculated and experimental data allowed the determination of the configuration of **1** as 19*S*,20-epoxy-18-oxotryprostatin A (Appendix A).

The HRESIMS spectrum of compound **2** exhibited an [M + Na]^+^ ion at *m/z* 436.1841 (calculated for C_22_H_27_N_3_O_5_Na, 436.1843), corresponding to a molecular formula of C_22_H_27_N_3_O_5_ (Appendix A). The UV spectrum of **2** showed maximal absorbance at 260 nm and 342 nm in MeOH. By comparing the ^1^H and 2D NMR data (Table 1, Appendix A) of **2** with those of **1**, it was evident that **2** possessed the same skeleton as **1**. The only differences included the absence of the epoxy moiety and the presence of one hydroxyl group. The HMBC correlation from H-19 (*δ*_H_ 3.05) to C-20 (*δ*_C_ 70.1), C-21 (*δ*_C_ 30.3), C-22 (*δ*_C_ 30.3), and C-18 (*δ*_C_ 194.1) indicated that the hydroxyl group was attached to C-20 and the methylene group (C-19) was attached to C-20 and C-18. On the basis of the above analysis, the structure of **2** was determined (Table 1 and Figure 3).

The structure of compound **4** can be found in the SciFinder database, however no literature and NMR data were reported for this compound. Here, we provide the NMR assignment of the compound. Compound **4** had the molecular formula C_18_H_18_O_10_S, as established from the [M − H]^−^ peak on HRESIMS (Appendix A) (*m/z* 425.0535 [M − H]^−^, calculated 425.0548). The ^13^C NMR spectrum showed 18 carbon signals. The ^1^H and ^13^C NMR spectra, together with 2D NMR (Table 2 and Appendix A) revealed the presence of four aromatic methine carbons (*δ*_C_/*δ*_H_ 112.4/7.43, d, *J* = 2.1 Hz; 108.3/7.06, d, *J* = 2.1 Hz; 110.2/6.39, s; 103.7/6.27, s); three methoxyl groups at *δ*_H_ 3.66 (*δ*_C_ 52.3, C-8), *δ*_H_ 3.65 (*δ*_C_ 56.2, C-9), and *δ*_H_ 3.31 (*δ*_C_ 55.9, C-7′); and one methyl group at 2.26 (*δ*_C_ 22.0, C-8′). In addition to the above carbon signals, there were ten sp^2^ hybridized quaternary carbons, including an α, β-unsaturated carbonyl group (C-10) and an ester group (C-7) at *δ*_C_ 199.1 and 165.5, four oxygenated aromatic carbons (*δ*_C_ 163.4, C-6′, *δ*_C_ 160.9, C-2′, *δ*_C_ 155.8, C-2, *δ*_C_ 154.1, C-4), and four aromatic carbons (*δ*_C_ 148.2, C-4′, *δ*_C_ 129.4, C-1, *δ*_C_ 127.1, C-6, *δ*_C_ 109.9, C-1′). These NMR data suggested that compound **4** could be a monomethylsulochrin derivative [23]. The molecular formula obtained from HRESIMS suggested the presence of one additional sulphate group. The sulphate group was determined to be attached to C-4 by comparing the NMR data with **6** (Table 2 and Figure 3).

The structure of compound **10** is given in one article, however, no NMR data were reported [24]. Here, we provide the NMR assignment of the compound. The molecular formula C_17_H_17_NO_7_ was assigned to **10** based on the HRESIMS molecular ion peak at *m/z* = 370.0895 ([M + Na]^+^, calculated 370.0897) (Appendix A). The ^13^C NMR spectrum showed 17 carbon signals. The ^1^H and ^13^C NMR spectra, together with 2D NMR (Table 3 and Appendix A) revealed the presence of a monosubstituted benzene ring (2H at *δ*_H_ 8.25, dd, *J* = 8.4 and 1.2 Hz; 2H at *δ*_H_ 7.52, dd, *J* = 8.4 and 7.4 Hz; 1H at *δ*_H_ 7.67, tt, *J* = 7.4 and 1.2 Hz), one oxymethine group at *δ*_H_ 4.39 (*δ*_C_ 74.9, C-9), one oxymethylene group at *δ*_H_ 4.42 (*δ*_C_ 57.1, C-10), two methyl groups: one oxygenated (*δ*_C_/*δ*_H_ 51.6/3.24) and one allylic (*δ*_C_/*δ*_H_ 5.2/1.65), and eight quaternary carbons at *δ*_C_ 196.6 (C-4), 196.4 (C-12), 186.0 (C-2), 166.4 (C-6), 133.4 (C-13), 110.1 (C-3), 92.5 (C-8), and 91.4 (C-5). Based on these characteristic structural features and a comprehensive database search, the compound was identified as a pseurotin A derivative, with a side chain replaced by a hydroxymethyl group [25] (Figure 3). The NMR data were assigned unambiguously (Table 3).

Other known compounds were identified as 18-oxotryprostatin A (**3**) [22], sulochrin (**5**) [26], monomethylsulochrin (**6**) [23], 6-methoxyspirotryprostatin B (**7**) [22], spirotryprostatin A (**8**) [27], spirotryprostatin C (**9**) [28], pseurotin A (**11**) [25], azaspirofuran A (**12**) [29], demethoxyfumitremorgin C (**13**) [30], fumitremorgin C (**14**) [28], 13-oxofumitremorgin B (**15**) [31], fumitremorgin B (**16**) [28], verruculogen (**17**) [28], helvolinic acid (**18**) [32], helvolic acid (**19**) [33], fumiquinazolines J (**20**) [34], chaetominine (**21**) [35], fumigaclavine C (**22**) [36], 9-deacetylfumigaclavine C (**23**) [36], fumagiringillin (**24**) [37], asterric acid (**25**), circinophoric acid (**26**) [38], dimethyl 2,3′-dimethylosoate (**27**) [23], methylated asterric acid (**28**) [39], endocrocin (**29**) [40], and questin (**30**) [41]. Their ^1^H and ^13^C NMR data were identical to those reported in the literature.

### 2.3. Biological Activities

All compounds were evaluated in vitro for antibacterial activities against *M. smegmatis*, *S. aureus*, *E. coli*, and *P. aeruginosa*, except for compounds **1**, **2**, and **4** because of limited amounts (Table 4). Compounds **15**, **16**, **20**, **23**, and **30** were active against certain test strains, showing weak activity with a shared minimum inhibitory concentration (MIC) value of 100 μg/mL. Compounds **18** and **19** exhibited strong activities against *S. aureus* (6.25 and 3.13 μg/mL, respectively) and *E. coli* (6.25 and 3.13 μg/mL, respectively). Other compounds were inactive at concentrations up to 100 μg/mL.

### 2.4. Proposed BGCs and Biosynthetic Pathway

The putative secondary metabolite BGCs of MF071 were predicted based on the antiSMASH results and a further detailed sequence analysis. We report here the BGCs for fumitremorgins (*ftm*), pseurotins (*pso*), fumigaclavines (*fga*), and helvolinic acid (*hel*) and associated biosynthetic pathways.

Fumitremorgins BGC *ftm* consists of eight genes, encoding one nonribosomal peptide synthetase, three cytochrome P450 monooxygenases, two prenyltransferases, one *O*-methyltransferase, and one oxygenase (Figure 4A, Appendix A). Based on the deduced function of each gene in predicted BGC (*ftm*, accession no. MT424560), the biosynthetic pathway of tryprostatins (**1**–**3**), spirotryprostatins (**7**–**9**), and fumitremorgins (**13**–**17**) was proposed with the initial catalysis of L-tryptophan, L-proline, and dimethylallyl diphosphate (Figure 4B) [42,43]. The catalytic evidence for the production of compounds **1**–**3**, **8**, **9**, and **15** are still to be uncovered.

When we analyzed the BGC of pseurotins from MF071, one PKS-NRPS hybrid cluster was found (*pso*, accession no. MT424563). Unexpectedly, the *pso* gene cluster intertwined with biosynthetic genes involved in the formation of fumagillins (Figure 5A) [44]. On the basis of the putative function of each gene through the BLASTp analysis (Appendix A), the biosynthetic pathway of pseurotins (**10**–**12**), one type of compound with an unusual heterospirocyclic γ-lactam feature, was proposed, starting with the condensation of one propionate (acetate for compound **10**), four malonates (two malonates for compound **10**), one L-methionine, and one L-phenylalanine (Figure 5) [45,46]. Notably, this is the first report of compound **10** as a natural product. Compared with the biosynthesis of most pseurotin derivatives, a different polyketide biosynthetic pathway was proposed for compound **10** (Figure 5C).

Bioinformatic analysis identified *fga* BGC (accession no. MT424562) as the putative biosynthetic cluster for fumigaclavines, consisting of 11 open reading frames (ORFs) and spanning 23 kb of genomic DNA (Figure 6A). The function of each gene was proposed by BLASTp analysis against the NCBI database using an amino acid sequence (Appendix A). The biosynthetic pathway of fumigaclavines (**22** and **23**) was also proposed (Figure 6). The enzymatic catalysis of prenylation at position C4 of indole by Fga3, methylation by Fga1, acetylation by Fga5, and prenylation at position 2 of indole by Fga8 have been proven by genetic approaches [47,48,49]. However, the formation of the D ring of the tetracyclic ergoline and the catalytic mechanism of the *tert*-prenylation at position C2 are still unclear.

A bioinformatic analysis and literature search also revealed the BGC of helvolinic acid (*hel*, accession no. MT424561) (Appendix A, Appendix A), and the biosynthetic pathway of helvolinic acid (**18**) and helvolic acid (**19**) was proposed (Appendix A) [50]. 

## 3. Discussion

Marine derived fungi are still important sources for the discovery of new bioactive natural products. The current study presents a genome-inspired metabolic mining of marine fungus *A. fumigatus* MF071, which led to the discovery of diverse BGCs and 30 compounds, including two new compounds and two known compounds with NMR data reported for the first time. Evaluation of antibacterial activity showed that compounds **18** and **19** exhibited strong activity against *S. aureus* and *E. coli*. A bioinformatic analysis of the genome sequences of MF071 revealed large numbers of secondary metabolite gene clusters. Careful inspection and analysis of the sequences revealed the BGCs for fumitremorgins (*ftm*), pseurotins (*pso*), fumigaclavines (*fga*), and helvolinic acid (*hel*). The putative BGC prediction fundamentally underpinned the enzymatic and mechanistic function for the biosynthesis of these compounds.

Fumitremorgins, pseurotins, and fumigaclavines were most frequently isolated from *A. fumigatus* strains. However, some of them were also reported from taxonomically close species, such as 18-oxotryprostatin A (**3**), 6-methoxyspirotryprostatin B (**7**) from *A. sydowi* [22], azaspirofuran A (**12**) from *A. sydowi* D2–6 [29], fumigaclavine I from *A. terreus* [51]. Helvolinic acid (**18**) was also isolated from *Corynascus setosus* and *M. anisopliae* [32,52]. Fumitremorgin B (**16**) showed weak activity against *M. smegmatis*, *S. aureus*, *E. coli*, *P. aeruginosa* in our in vitro assay. It was also reported with antifungal activity against a variety of phytopathogenic fungi, which could be involved in fighting against invasion by other pathogens [53]. Our research also showed the strong activities of helvolinic acid (**18**) and helvolic acid (**19**) against *S. aureus* (6.25 and 3.13 μg/mL, respectively) and *E. coli* (6.25 and 3.13 μg/mL, respectively). Previous studies revealed that helvolic acid (**19**) exhibited in vitro antimycobacterial activity against *M. tuberculosis* H37Ra [54], antitrypanosomal activity against *Trypanosoma brucei brucei* [55], and antimalarial activity against multidrug resistant *Plasmodium falciparum* [56]. No cytotoxic activity against normal cell lines and broad biological activity indicated the potential of helvolic acid for drug development [56].

Pseurotins have a unique heterospirocyclic furanone-lactam structure. They are produced by hybrid PKS/NRPS and other tailing enzymes, and exhibit a broad range of biological activities. However, the compounds showed no antibacterial activity in our screening at concentrations up to 100 μg/mL, which did not agree with the results of antibacterial activity against *E. coli* (ATCC 25922), *P. aeruginosa* (ATCC 27853), *S. aureus* (ATCC 25923) from Pinheiro et al. [57] It is likely that different bacterial strains contributed to the different results, as pseurotin A was also reported to have no activity against *S. aureus* (ATCC 6538) and *S. aureus* [58,59]. The mechanism of the biosynthesis of the unusual spiro-ring structural feature of pseurotins has remained uncharacterized. We propose that it could be formed by isomerization and hydroxylation. Previous research showed the physically intertwined supercluster genes for the biosynthesis of both pseurotin A and fumagillin. The gene *fumR* regulates the production of pseurotin A and fumagillin. It was intriguing that the presence of genes in the cluster which were similar to fumagillin targets conferred the strain resistance to fumagillin [60]. However, fumagillin was not isolated from the extract of MF071, possibly due to the low yield.

Several putative prenyltransferases were identified in MF071 for the incorporation of one prenyl moiety in the biosynthesis of fumitremorgins and fumigaclavines, such as Ftm4, Ftm8, Fga3, and Fga8. Fga8 catalyzed a “reverse” prenylation of fumigaclavine A with the 2-(1,1-dimethylallyl) moiety connected to the indole system at the 2-position (Figure 6B). The whole genome sequence analysis of MF071 revealed two additional prenyltransferases. Gene deletion experiments or heterologous expression have revealed the function of most “reverse” prenyltransferase genes, such as *lxc* from *Lyngbya majuscule*, *notF* from *Aspergillus* sp., *anaPT* from *Neosartorya fischeri*, *brePT* from *Aspergillus versicolor*, and *cdpC2PT*/*cdpNPT* from *A. nidulans* [61,62,63,64,65]. However, the mechanism of the enzymatic catalysis of these “reverse” prenyltransferases has not been fully revealed. The amino acid sequence alignment of Fga8 with the above-mentioned prenyltransferases gave a relatively low similarity value (20%-30%), indicating that Fga8 could be a potential new prenyltransferase. A possible mechanism of the *tert*-prenylation at position C2 by Fga 8 was that the prenyltransferase Fga8 initially alkylates the nitrogen atom of the indole. The resulting *N*-(3,3-dimethylallyl) indole then undergoes an aza-Claisen rearrangement to give the rearranged 3-(1,1-dimethylallyl)indole, followed by a [1,5]-alkyl shift and aromatization to give the corresponding 2-substituted indole (Figure 7). 

We require more information to test the substrate specificity of these prenyltransferases, as prenylations or *tert*-prenylations of indole could occur at positions N1, C2, C3, C4, C5, C6, and C7 (Appendix A). Interestingly, the brevianamide F could be catalyzed by both FtmB (“regular” prenyltransferase) and NotF to produce tryprostatin B and deoxybrevianamide E, respectively [42,62]. CdpC3PT from *A. nidulans* has been reported to catalyse the formation of N1-regularly, C2-, and C3- reverse-prenylated derivatives [65]. Further protein structure research could be of importance to confirm this prenylation mechanism.

The current study reports 30 compounds and BGCs of fumitremorgins, pseurotins, fumigaclavines and helvolinic acid, whereas the prediction of MF071 metabolic potential gave large numbers of BGCs. A preliminary blast analysis showed the presence of BGCs for pyripyropene A, neosartoricin B, gliotoxin, trypacidin, xanthocillin, fumisoquin, ferricrocin, 1,8-dihydroxynaphthalene, and many others. However, biosynthetic genes are often silent or transcribed at very low levels under certain conditions, which makes the detection difficult. As the condition used for fermentation is quite different from the native environment (high salinity, oligotrophy, microbial competition, temperature variation, etc.), the chemical profile could be different from that of the extract from rice medium fermentation. Approaches for the activation of these silent BGCs such as OSMAC, microbial co-culture or heterologous expression of unknown clusters could be carried out to further expand the structure classes.

In conclusion, we isolated 30 compounds from *A. fumigatus* MF071, including two new compounds **1** and **2**. The NMR data of two compounds, monomethylsulochrin-4-sulphate (**4**) and pseurotin H (**10**), are also reported here for the first time. Compounds **18** and **19** exhibited strong activities against *S. aureus* and *E. coli*. BGCs of fumitremorgins, pseurotins, fumigaclavines and helvolinic acid and biosynthetic pathways were proposed. The mechanism for the *tert*-prenylation of indoles by prenyltransferase was also discussed.

## 4. Materials and Methods

### 4.1. General Experimental Procedures

NMR spectra were acquired at 25 °C on a Bruker Avance HDX 800 MHz spectrometer (Zürich, Switzerland) equipped with a TCI cryoprobe. The ^1^H and ^13^C chemical shifts were referenced to the DMSO-*d*6 solvent peaks at *δ*_H_ 2.50 and *δ*_C_ 39.52 ppm, respectively, and all deuterated solvents were from Cambridge Isotope Laboratories (CIL). Low resolution mass spectra were measured with a Thermo Ultimate 3000 system equipped with an Accucore^TM^ C18 column (2.6 μm, 150 × 2.1 mm), a diode-array detector (DAD), and an ESI mass spectrometer. HRESIMS measurements were obtained on a Bruker Maxis II ETD QTOF mass spectrometer (Bremen, Germany) and was calibrated with sodium trifluoroacetate. SINGLE StEP Silica Column™ and Sephadex LH-20 (GE Healthcare BioSciences AB) were used for fractionation. Reverse phase HPLC was performed on Thermo Ultimate 3000 system separation module with a Dionex^TM^ diode array detector (MA, USA). Optical rotations were determined on a Jasco P-1020 Polarimeter (10 cm cell) (Tokyo, Japan). All solvents used for extraction, chromatography, [α]_D_, and MS were Honeywell Burdick & Jackson HPLC grade (Muskegon, MI, USA), 0.1% formic acid (Sigma-Aldrich) was used in solvent system for LC-MS and 0.1% TFA (Sigma-Aldrich) was used in solvent system for HPLC. H_2_O was purified with Sartorius Arium^®^Pro VF ultrapure water system (Göttingen, Germany).

### 4.2. Microbial Strain Culture and Identification

The marine fungus *Aspergillus fumigatus* MF071 (MF071) was isolated from a sediment sample collected at a depth of 60 m from the Bohai Sea, China. Specifically, 1.0 g of sediment sample was added into 50 mL sterile centrifuge tube and suspended in 9 mL sterile artificial seawater (3.8% sea salt) under aseptic operation. An aliquot of 200 µL diluted suspension (1/10) was spread plated on the separation medium (1.0% peptone, 4.0% glucose, 1.5% agar, pH 6.0), supplemented with 0.5 mg/mL chloramphenicol and streptomycin, and 200 µL sterile artificial seawater was also spread, plated on another plate as control. The plate was incubated at 28 °C. The pure colony of MF071 was transferred to potato dextrose agar (PDA) medium for further lab experiments and cryogenic vials, with MF071 suspended in 25% glycerol stored at −80 °C. A DNA extraction of MF071 was carried out using CTAB (cetyltrimethylammonium bromide) as described previously [66]. The identification of strain MF071 was performed based on the morphological and 18S ribosomal DNA (rDNA) analyses. Multiple sequence alignments with 18S sequences of related species were carried out using CLUSTAL W [67]. A phylogenetic tree was constructed using the neighbor-joining method [68], as implemented in MEGA 5.0 [69]. Bootstrap values were generated by resampling 1000 replicates. The voucher specimen has been deposited at Dr. Zhang’s Laboratory, East China University of Science and Technology (strain no. MF071). 

### 4.3. Genome Sequencing and Secondary Metabolite BGCs Analysis

Whole-genome sequencing of strain MF071 was conducted on PacBio RSII platform (Tianjin Biochip Corporation, Tianjin, China), with the single molecule real-time (SMRT) technique [70]. The data from a single SMRT sequencing cell were used directly for the assembly process. The raw reads were processed and assembled using the hierarchical genome assembly process (HGAP) to obtain the final genomic sequence [71]. Gene prediction of the draft genome assembly was performed using AUGUSTUS [72]. The prediction of secondary metabolite BGCs was carried out using antiSMASH online software (version 5.1.2) [73]. Each gene in the putative gene clusters was analyzed by BLASTp against the GenBank database [74].

### 4.4. Fermentation, Extraction, and Isolation

Strain MF071 was cultured on a PDA at 28 °C for seven days, and agar plugs (5-mm-diameter) were aseptically inoculated into three conical flasks (250 mL), each containing 50 mL of potato dextrose broth (PDB). The flasks were incubated at 28 °C on a rotary shaker at 200 rpm for five days to generate the seed cultures, which were distributed into 20 conical flasks (1000 mL), each containing 160 g of autoclaved rice and 240 mL distilled H_2_O. These cultures were statically fermented at 28 °C for thirty days. The fermentation products were extracted with EtOAc three times and concentrated in vacuo to give a crude extract (20 g).

A MF071 extract was fractionated on a Sephadex LH-20 column using MeOH and DCM (1:1), affording 11 fractions (F1-F11). F8 (20 mg) was purified by semi-preparative RP-HPLC using an Alltech Hyperprep C18 column (5 μm, 250 × 21.2 mm), eluting at a flow rate of 9.0 mL/min with a gradient elution of MeOH-H_2_O: 0-50 min, 10%/90% to 100%/0%, to obtain compound **5** (0.9 mg, *t*_R_ = 21.5 min). F9 (4.8 mg) was subjected to Sephadex LH-20 column to give compound **4**. F10 (23 mg) was purified by semi-preparative RP-HPLC on a Phenomenex Luna C18 column (5 μm, 250 × 10 mm), eluting at a flow rate of 4.0 mL/min with a gradient elution from 10% MeOH to 100% MeOH in 50 min, to obtain sub-fraction 20 (F20), compound **29** (1.4 mg, *t*_R_ = 31.2 min), and **30** (1.3 mg, *t*_R_ = 37.0 min). F20 was further purified using RP-HPLC, with a gradient elution from 40% MeOH to 65% MeOH in 25 min, to obtain compound **6** (0.1 mg, *t*_R_ = 23.0 min).

F4 (6.2 g) was subjected to Single StEP Silica Column™, and eluted with gradient DCM and MeOH to afford 8 sub-fractions (F4A-F4H). F4D (280 mg) was purified using RP-HPLC on a Thermo Electron Betasil C18 column (5 μm, 150 × 21.2 mm), eluting at a flow rate of 9.0 mL/min, with a gradient elution from 10% MeOH to 100% MeOH in 50 min, to give fraction 35. Fraction 35 was further purified on a Phenomenex Luna C18 column (5 μm, 250 × 10 mm), eluting at a flow rate of 4.0 mL/min with 60% MeOH to obtain compound **25** (*t*_R_ = 12.0 min). F4B was separated using RP-HPLC on a Thermo Electron Betasil C18 column (5 μm, 150 × 21.2 mm), eluting at a flow rate of 9.0 mL/min with a gradient elution from 10% MeOH to 100% MeOH in 50 min, to give six sub-fractions (F4B1- F4B6). 

F4B2 (284 mg) was purified using RP-HPLC on a Phenomenex Luna C18 column (5 μm, 250 × 21.2 mm), eluting at a flow rate of 9.0 mL/min, with a gradient elution from 10% MeOH to 100% MeOH in 50 min, to give sub-fractions F4B2_F20 and F4B2_F27. F4B2_F20 was further purified using RP-HPLC on a Phenomenex Luna C18 column (5 μm, 250 × 10 mm), eluting at a flow rate of 4.0 mL/min, with 30% MeOH, to give compound **10** (0.9 mg, *t*_R_ = 16.7 min) and **11** (*t*_R_ = 30.2 min).

F4B3 (619 mg) was purified using RP-HPLC on a Phenomenex Luna C18 column (5 μm, 250 × 21.2 mm), eluting at a flow rate of 9.0 mL/min, with a gradient elution from 30% MeOH to 100% MeOH in 50 min, to give sub-fractions F4B3_F19, F4B3_F23 and F4B3_F28. F4B3_F19 was further purified using RP-HPLC on a Phenomenex Luna C18 column (5 μm, 250 × 10 mm), eluting at a flow rate of 4.0 mL/min, with a gradient 30% MeOH to 62% MeOH in 25 min, to give compound **26** (1.6 mg, *t*_R_ = 22.2 min). F4B3_F23 was further purified using RP-HPLC on a Phenomenex Luna C18 column (5 μm, 250 × 10 mm), eluting at a flow rate of 4.0 mL/min, with 50% MeOH, to give compound **21** (2.7 mg, *t*_R_ = 18.8 min). F4B3_F28 was further purified using RP-HPLC on a Phenomenex Luna C18 column (5 μm, 250 × 10 mm), eluting at a flow rate of 4.0 mL/min, with 45% MeOH, to give compound **8** (0.1 mg, *t*_R_ = 9.7 min), **2** (*t*_R_ = 11.9 min), **1** (*t*_R_ = 12.7 min), **12** (0.1 mg, *t*_R_ = 14.5 min), **28** (1.0 mg, *t*_R_ = 16.4 min), **3** (1.8 mg, *t*_R_ = 18.8 min).

F4B4 (373 mg) was purified using RP-HPLC on a Phenomenex Luna C18 column (5 μm, 250 × 21.2 mm), eluting at a flow rate of 9.0 mL/min, with a gradient elution from 30% MeOH, to 100% MeOH in 50 min to give sub-fractions F4B4_F30, F4B4_F34, F4B4_F39 and F4B4_F40. F4B4_F30 was further purified using RP-HPLC on a Phenomenex Luna C18 column (5 μm, 250 × 10 mm), eluting at a flow rate of 4.0 mL/min, with 45% MeOH, to give compound **7** (*t*_R_ = 10.7 min). F4B4_F34 was further purified using RP-HPLC on a Phenomenex Luna C18 column (5 μm, 250 × 10 mm), eluting at a flow rate of 4.0 mL/min, with a gradient from 30% MeOH to 70% MeOH in 20 min, to give compound **23** (*t*_R_ = 14.7 min) and **22** (*t*_R_ = 17.1 min). F4B4_F39 was further purified using RP-HPLC on a Phenomenex Luna C18 column (5 μm, 250 × 10 mm), eluting at a flow rate of 4.0 mL/min, with 65% MeOH, to give compound **9** (*t*_R_ = 17.5 min) and **24** (*t*_R_ = 20.5 min). F4B4_F40 was further fractionated using Sephadex LH-20 to give compound **20**.

F4B5 (355 mg) was subjected on a Sephadex LH-20 column using MeOH and DCM (1:1). Sub-fraction 26 was purified using RP-HPLC on a Phenomenex Luna C18 column (5 μm, 250 × 10 mm), eluting at a flow rate of 4.0 mL/min, with a gradient from 50% MeOH to 100% MeOH in 25 min, to give compound **15** (0.4 mg, *t*_R_ = 16.1 min) and **19** (2.6 mg, *t*_R_ = 19.9 min). Sub-fraction 33 was purified using RP-HPLC on a Phenomenex Luna C18 column (5 μm, 250 × 10 mm), eluting at a flow rate of 4.0 mL/min, with 45% MeOH, to give compound **27** (2.5 mg, *t*_R_ = 5.1 min), **14** (1.0 mg, *t*_R_ = 7.0 min), **13** (0.5 mg, *t*_R_ = 7.5 min), **17** (1.1 mg, *t*_R_ = 10.8 min), **18** (0.5 mg, *t*_R_ = 12.5 min), and **16** (3.9 mg, *t*_R_ = 13.1 min).

Compound **1**: yellow powder; UV (MeOH) *λ*_max_ 260 and 350 nm; ^1^H and ^13^C NMR data, see Table 1; 2D NMR spectra, see Appendix A; HRESIMS *m/z* 434.1689 [M + Na]^+^ (calculated for 434.1686) (Appendix A).

Compound **2**: yellow powder; UV (MeOH) *λ*_max_ 260 and 342 nm; ^1^H and ^13^C NMR data, see Table 1; 2D NMR spectra, see Appendix A; HRESIMS *m/z* 436.1841 [M + Na]^+^ (calculated for 436.1843) (Appendix A).

Compound **4**: yellow powder; UV (MeOH) *λ*_max_ 291 nm; ^1^H and ^13^C NMR data, see Table 2; 2D NMR spectra, see Appendix A; HRESIMS *m/z* 425.0535 [M − H]^−^ (calculated for 425.0548) (Appendix A).

Compound **10**: yellow powder; UV (MeOH) *λ*_max_ 260 nm; ^1^H and ^13^C NMR data, see Table 1; 2D NMR spectra, see Appendix A; HRESIMS *m/z* 370.0895 [M + Na]^+^ (calculated for 370.0897) (Appendix A). 

### 4.5. DFT Theory and Calculation

The NMR chemical shift calculations were performed using density functional theory (DFT) in Gaussian 09. The preliminary conformational distribution search was performed by HyperChem Release 8.0 software. GaussView 5.0 was used to view the conformational structures and change the input file for calculation. All ground-state geometries were optimized at the B3LYP/6-31G(d) level, and the stable conformations obtained at the B3LYP/6-31G(d) level were further used in magnetic shielding constants at the B3LYP/6-311++G(2d,p) level [75]. The calculated chemical shift of each atom in each conformer was viewed by GaussView 5.0, and the final chemical shift of each atom was calculated from the Boltzmann distribution of each conformer. A DP4 probability analysis of the calculated and experimental chemical shifts was used to assign the stereochemistry [76].

### 4.6. Bioassays 

The bioactivity of isolated compounds was tested against *Mycobacterium smegmatis* (ATCC 70084), *Staphylococcus aureus* (ATCC BAA-2312), *Escherichia coli* (ATCC 43887), and *Pseudomonas aeruginosa* (ATCC 10145), using a 96-well plate microdilution method, as previously reported [77,78,79]. The minimum inhibitory concentrations (MICs) were calculated as the minimum concentration of the compounds that inhibited visible growth. Isoniazid was used as a positive control in the activity screening against *M. smegmatis* with MIC value of 4 μg/mL. Gentamycin was used as a positive control in the activity screening against *S. aureus*, *E. coli*, and *P. aeruginosa*, with MIC values of 0.5 μg/mL, 0.03 μg/mL, 1.0 μg/mL, respectively. All the experiments were performed in triplicate.

## Figures and Tables

**Figure 1 marinedrugs-18-00352-f001:**
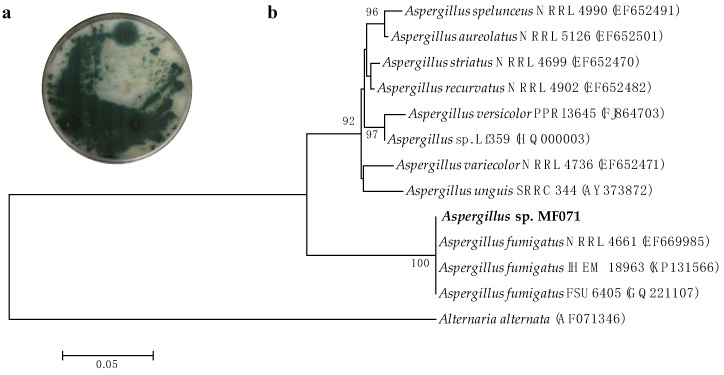
Morphology and phylogenetic tree of *Aspergillus fumigatus* MF071. (**a**) Colony characteristics of *A. fumigatus* MF071 grown on potato dextrose agar at 28 °C for 10 days; (**b**) Neighbor-joining tree of *A. fumigatus* MF071 based on 18S sequences. Numbers at nodes indicate levels of bootstrap support (%) based on a neighbor-joining analysis of 1000 resampled datasets; only values >50 % are shown. National Center for Biotechnology Information (NCBI) accession numbers are provided in parentheses. The Bar represents 0.05 nucleotide substitutions per site.

**Figure 2 marinedrugs-18-00352-f002:**
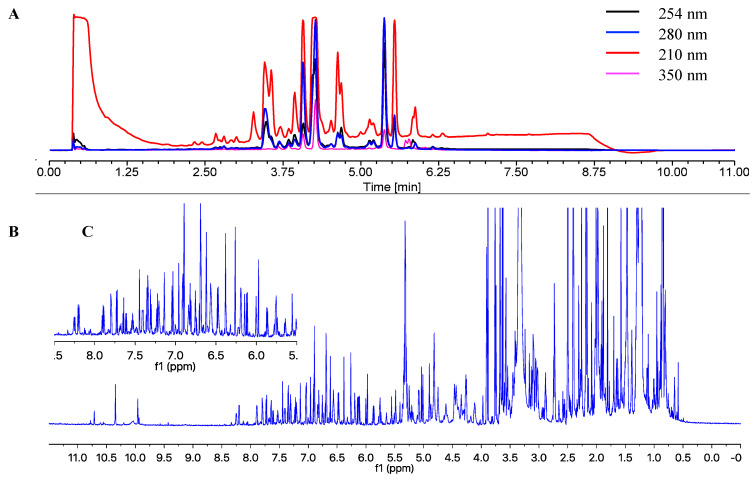
HPLC spectrum (**A**), ^1^H NMR spectrum (**B**), and ^1^H NMR expansion of MF071 extract.

**Figure 3 marinedrugs-18-00352-f003:**
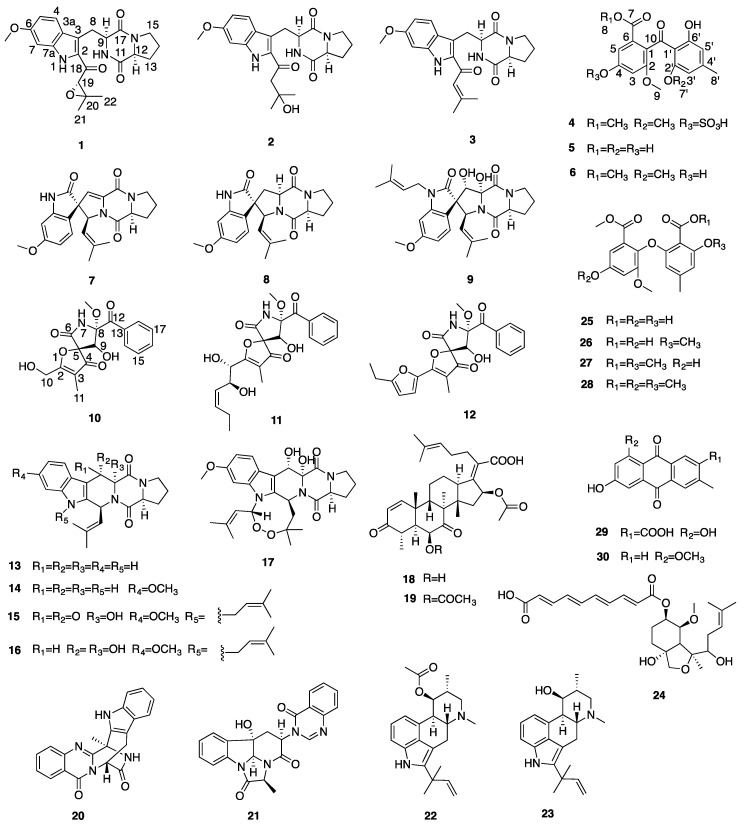
Structures of compounds isolated from *Aspergillus fumigatus* MF071.

**Figure 4 marinedrugs-18-00352-f004:**
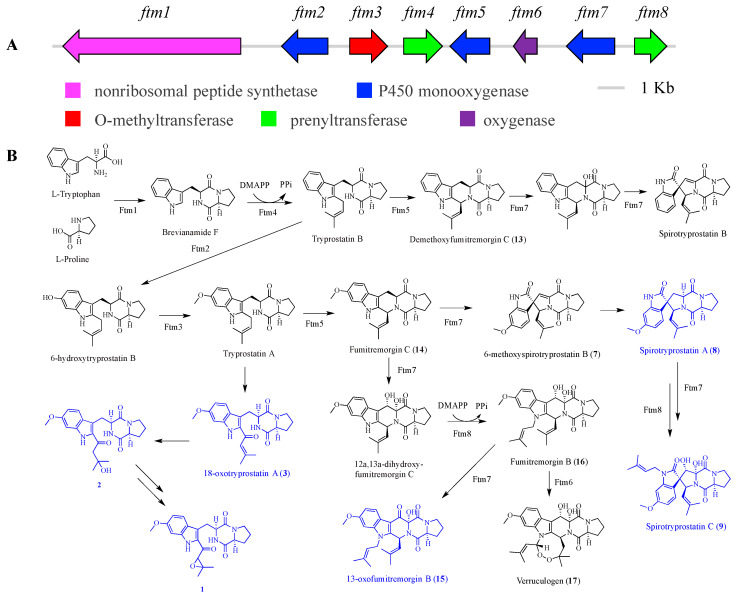
Organization of the fumitremorgin-type indole alkaloids BGC (*ftm*) (**A**), and proposed biosynthetic pathways for tryprostatins, spirotryprostatins, and fumitremorgins (**B**).

**Figure 5 marinedrugs-18-00352-f005:**
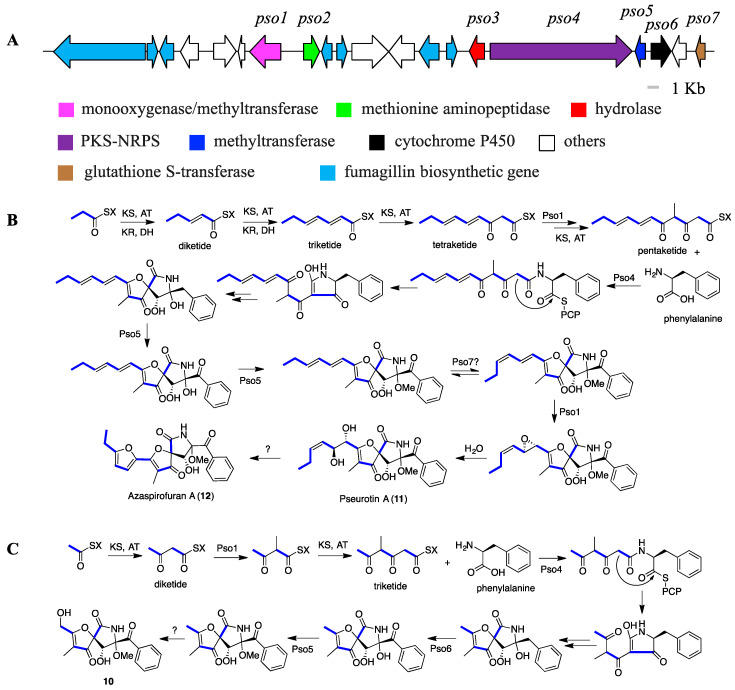
Organization of the pseurotins biosynthetic gene clusters (BGC) (*pso*) (**A**), and proposed biosynthetic pathways for pseurotin A and azaspirofuran A (**B**), and compound **10** (**C**).

**Figure 6 marinedrugs-18-00352-f006:**
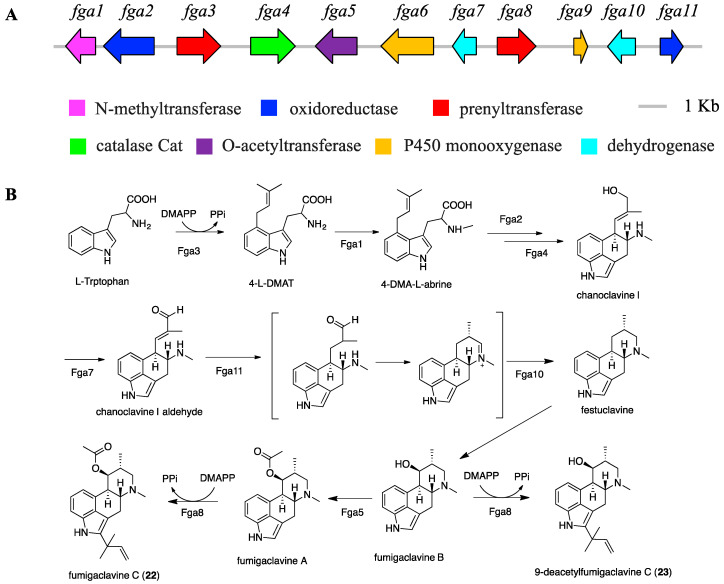
Organization of the fumigaclavines BGC (*fga*) (**A**), and proposed biosynthetic pathway for fumigaclavine C and 9-deacetylfumigaclavine C (**B**).

**Figure 7 marinedrugs-18-00352-f007:**
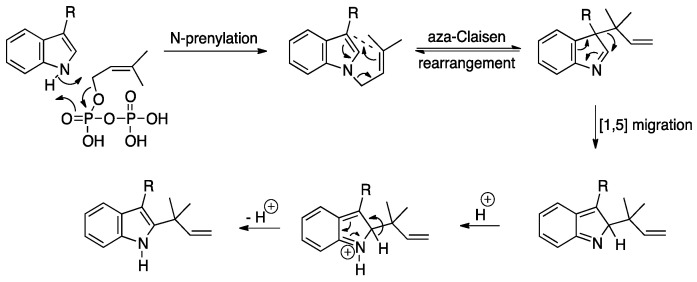
Proposed mechanism for the *tert*-prenylation of indoles at the C-2 position.

**Table 1 marinedrugs-18-00352-t001:** ^1^H (800 MHz) and ^13^C (200 MHz) NMR data of compounds **1** and **2.**

Position	1		2	
	*δ*_C_, Type	*δ*_H_, Mult (*J* in Hz)	*δ*_C_, Type	*δ*_H_, Mult (*J* in Hz)
1		11.57 s		
2	131.7, C		133.8, C	
3	119.8, C		119.2, C	
3a	122.4, C		122.3, C	
4	123.0, CH	7.68 d (8.9)	122.7, CH	7.64 d (8.9)
5	112.5, CH	6.77 dd (8.9, 2.3)	112.2, CH	6.74 dd (8.9, 2.3)
6	159.8, C		159.3, C	
7	94.1, CH	6.89 d (2.3)	94.1, CH	6.88 d (2.3)
7a	138.7, C		138.0, C	
8	26.0, CH_2_	3.62 m, 3.32 m	25.5, CH_2_	3.61 dd (14.2, 4.9), 3.27 m
9	56.1, CH	4.42 t (6.5)	56.6, CH	4.36 t (6.1)
10		7.45 s		7.42 s
11	167.2, C		167.0, C	
12	58.9, CH	4.16 t (8.0)	58.8, CH	4.14 t (8.0)
13	28.1, CH_2_	2.07 m, 1.72 m	28.0, CH_2_	2.06 m, 1.73 m
14	22.6, CH_2_	1.75 m	22.7, CH_2_	1.75 m
15	45.5, CH_2_	3.39 m, 3.28 m	45.3, CH_2_	3.38 m, 3.32 m
17	165.9, C		166.0, C	
18	187.2, C		194.1, C	
19	63.9, CH	4.36 s	52.9, CH_2_	3.05 d (13.9), 3.02 d (13.9)
20	61.8, C		70.1, C	
21	24.7, CH_3_	1.48 s	30.3, CH_3_	1.25 s
22	18.6, CH_3_	1.17 s	30.3, CH_3_	1.25 s
6-OCH_3_	55.6, CH_3_	3.81 s	55.6, CH_3_	3.80 s

**Table 2 marinedrugs-18-00352-t002:** ^1^H (800 MHz) and ^13^C (200 MHz) NMR data of compounds **4** and **6.**

Position	4		6	
	*δ*_C_, Type	*δ*_H_, Mult (*J* in Hz)	*δ*_C_, Type	*δ*_H_, Mult (*J* in Hz)
1	129.4, C		125.8, C	
2	155.8, C		156.6, C	
3	108.3, CH	7.06 d (2.1)	103.2, CH	6.69 d (2.2)
4	154.1, C		158.1, C	
5	112.4, CH	7.43 d (2.1)	107.2, CH	6.89 d (2.2)
6	127.1, C		128.0, C	
7	165.5, C		165.8, C	
8	52.3, CH_3_	3.66 s	52.1, CH_3_	3.62 s
9	56.2, CH_3_	3.65 s	56.0, CH_3_	3.63 s
10	199.1, C		199.4, C	
1’	109.9, C		110.1, C	
2’	160.9, C		160.8, C	
3’	103.7, CH	6.27 s	103.5, CH	6.26 s
4’	148.2, C		147.8, C	
5’	110.2, CH	6.39 s	110.1, CH	6.38 s
6’	163.4, C		163.3, C	
7’	55.9, CH_3_	3.31 s	55.9, CH_3_	3.33 s
8’	22.0, CH_3_	2.26 s	21.9, CH_3_	2.26 s
6′-OH		12.90 s		12.95 s
4-OH				10.05 s

**Table 3 marinedrugs-18-00352-t003:** ^1^H (800 MHz) and ^13^C (200 MHz) NMR data of compound **10.**

Position	*δ*_C_, Type	*δ*_H_, Mult (*J* in Hz)	Position	*δ*_C_, Type	*δ*_H_, Mult (*J* in Hz)
1			11	5.2, CH_3_	1.65 s
2	186.0, C		12	196.4, C	
3	110.1, C		13	133.4, C	
4	196.6, C		14	130.3, CH	8.25 dd (8.4, 1.2)
5	91.4, C		15	128.4, CH	7.52 dd (8.4, 7.4)
6	166.4, C		16	133.8, CH	7.67 tt (7.4, 1.2)
7		9.92 s	17	128.4, CH	7.52 dd (8.4, 7.4)
8	92.5, C		18	130.3, CH	8.25 dd (8.4, 1.2)
9	74.9, CH	4.39 s	8-OCH_3_	51.6, CH_3_	3.24 s
10	57.1, CH_2_	4.42 d (3.0)			

**Table 4 marinedrugs-18-00352-t004:** Antibacterial activity of identified compounds.

Compounds	Pathogenic Bacteria (MIC, μg/mL)			
	*M. smegmatis*	*S. aureus*	*E. coli*	*P. aeruginosa*
**15**	>100	100	>100	>100
**16**	100	100	100	100
**18**	>100	6.25	6.25	>100
**19**	100	3.13	3.13	>100
**20**	100	100	>100	>100
**23**	100	>100	>100	>100
**30**	>100	100	100	>100

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
