# Peer review of "Genome-Inspired Chemical Exploration of Marine Fungus Aspergillus fumigatus MF071"

_marinedrugs, 2020, doi:10.3390/md18070352_

Round 1

Reviewer 1 Report

The study Genome-inspired chemical exploring of marine fungus Aspergillus fumigatus MF071 carried out by Han, Miaomiao, Jenkins, Liu, Zhang, Quinn, and Feng, reports 30 compounds and 4 biosynthetic gene clusters, identified in an Aspergillus strain isolated from the sea. Moreover, they predicted the metabolic potential of the strain, giving a large numbers of BGCs.

The study is well presentented and interesting for the scientific community as 2 new compounds have been described. Moreover, helvolic acid has been detected, with high antibacterial effect.

However, there are some general points that could be improved:

  • The isolation procedure must be better explained to assure that this strain is not an environmental contamination. Controls?
  • It would be intersesting to discuss if this marine strain has differences with the environmental non-marine or clinical strains. I mean, the authors should explain if BCGs described are also usually present in non-marine strains. It would be interesting to know if there is any result obtained thanks to that it is marine.
  • It would be interesting to include any bacterial isolate resistant to drugs, for example a methicillin resistant strain,to carry out MICs.
  • In opinion of this revieweer, discussion section must be rewriten to discuss more about the importance of the compounds found and their relevance for microbiology. For some of them, at least the most relevant, it should be discussed about their production by other fungal species, the effect that has been described for them in bibliography... It is very surprising, for example, that, In all the discussion there are not any explanation or discussion about the helvolic acid, which seems to be relevant in this study
  • In the same way, the discussion should include a conclusion paragraph, where the reader can find a summary of the results that you want to highlight and the main conclusions obtained.
  • Abstract: Don´t use abreviations (for example BCG)
  • Abstract: It should be included a conclusion sentence to summarize the most important findings.
  • In the text there is not a correct use of the BCG abreviation as sometimes it is abreviated and sometimes it is not (for example page 2, line 59)
  • There is a recent review about fumagillin, which should be included to explain de intertwin between pseurotin and fumagillin gene clusters and to explain its antimicrobial effect (Guruceaga et al., 2020.Fumagillin, a Mycotoxin of Aspergillus fumigatus:Biosynthesis, Biological Activities, Detection, and Applications. Toxins. 12, 7; doi:10.3390/toxins12010007)

Author Response

Reviewer 1:

The study Genome-inspired chemical exploring of marine fungus Aspergillus fumigatus MF071 carried out by Han, Miaomiao, Jenkins, Liu, Zhang, Quinn, and Feng, reports 30 compounds and 4 biosynthetic gene clusters, identified in an Aspergillus strain isolated from the sea. Moreover, they predicted the metabolic potential of the strain, giving a large numbers of BGCs.

The study is well presented and interesting for the scientific community as 2 new compounds have been described. Moreover, helvolic acid has been detected, with high antibacterial effect.

However, there are some general points that could be improved:

  1. The isolation procedure must be better explained to assure that this strain is not an environmental contamination. Controls?

A detailed description has been included in 4.2 Microbial Strain Culture in the manuscript (line 341-347), explaining the procedure of the strain isolation:

“Specifically, 1.0 g of sediment sample was added into 50 mL sterile centrifuge tube and suspended in 9 mL sterile artificial seawater (3.8% sea salt) under aseptic operation. An aliquot of 200 µL diluted suspension (1/10) was spread plated on the separation medium (1.0% peptone, 4.0% glucose, 1.5% agar, pH 6.0) supplemented with 0.5 mg/mL chloramphenicol and streptomycin. The plate was incubated at 28°C. The pure colony of MF071 was transferred to potato dextrose agar (PDA) medium for further lab experiments and cryogenic vials with MF071 suspended in 25% glycerol were stored at -80°C.”

  1. It would be interesting to discuss if this marine strain has differences with the environmental non-marine or clinical strains. I mean, the authors should explain if BGCs described are also usually present in non-marine strains. It would be interesting to know if there is any result obtained thanks to that it is marine.

Authors would like to thank reviewer’s valuable suggestion. However, our research didn’t reveal obvious differences between marine and non-marine strains in terms of the BGCs. Sequence similarity analysis of each BGCs showed that all four BGCs were present in marine and non-marine strains (a patient derived A. fumigatus Af293, clinical isolate A. fumigatus A1163), and they showed high sequence similarity (supporting information table S2-S5).

  1. It would be interesting to include any bacterial isolate resistant to drugs, for example a methicillin resistant strain, to carry out MICs.

We agree with the reviewer’s view on testing a methicillin resistant strain, this will form part of our future work.

  1. In opinion of this revieweer, discussion section must be rewritten to discuss more about the importance of the compounds found and their relevance for microbiology. For some of them, at least the most relevant, it should be discussed about their production by other fungal species, the effect that has been described for them in bibliography... It is very surprising, for example, that, In all the discussion there are not any explanation or discussion about the helvolic acid, which seems to be relevant in this study.

We have taken on board the reviewer’s suggestion and substantially expanded the discussion on the production of fumitremorgins, pseurotins, fumigaclavines and helvolinic acid, and the activity of each structure class. The revised discussion is as following with the new addition highlighted (line 239-line 321):

“Marine derived fungi are still important sources for the discovery of new bioactive natural products. The current study presented a genome-inspired metabolic mining of marine fungus A. fumigatus MF071, which led to the discovery of diverse BGCs and 30 compounds, including two new compounds and two known compounds with NMR data reported for the first time. Evaluation of antibacterial activity showed that compounds 18 and 19 exhibited strong activity against S. aureus and E. coli. Bioinformatic analysis of the genome sequences of MF071 revealed large numbers of secondary metabolite gene clusters. Careful inspection and analysis of the sequences revealed the BGCs for fumitremorgins (ftm), pseurotins (pso), fumigaclavines (fga), and helvolinic acid (hel). The putative BGC prediction fundamentally underpinned the enzymatic and mechanistic function for the biosynthesis of these compounds.

Fumitremorgins, pseurotins, and fumigaclavines were most frequently isolated from A. fumigatus strains. However, some of them were also reported from taxonomically close species. such as 18-oxotryprostatin A (3), 6-methoxyspirotryprostatin B (7) from A. sydowi [22], azaspirofuran A (12) from A. sydowi D2–6 [29], fumigaclavine I from A. terreus [51]. Helvolinic acid (18) was also isolated from Corynascus setosus and M. anisopliae [32,52]. Fumitremorgin B (16) showed weak activity against M. smegmatis, S. aureus, E. coli, P. aeruginosa in our in vitro assay. It was also reported with antifungal activity against a variety of phytopathogenic fungi, which could be involved in fighting against invasion by other pathogens [53]. Our research also showed the strong activities of helvolinic acid (18) and helvolic acid (19) against S. aureus (6.25 and 3.13 μg/mL, respectively) and E. coli (6.25 and 3.13 μg/mL, respectively). Previous studies revealed that helvolic acid (19) exhibited in vitro antimycobacterial activity against M. tuberculosis H37Ra [54], antitrypanosomal activity against Trypanosoma brucei brucei [55], and antimalarial activity against multidrug resistant Plasmodium falciparum [56]. No cytotoxic activity against normal cell lines and broad biological activity indicated the potential of helvolic acid for drug development [56].

Pseurotins had a unique heterospirocyclic furanone-lactam structure. They were produced by hybrid PKS/NRPS and other tailing enzymes, and exhibited a broad range of biological activities. However, the compounds showed no antibacterial activity in our screening at concentrations up to 100 μg/mL, which did not agree with the results of antibacterial activity against E. coli (ATCC 25922), P. aeruginosa (ATCC 27853), S. aureus (ATCC 25923) from Pinheiro et al [57]. It is likely that different bacterial strains contributed to the different results as pseurotin A was reported to have no activity against S. aureus (ATCC 6538) and S. aureus [58,59]. The mechanism of biosynthesis of the unusual spiro-ring structural feature of pseurotins has remained uncharacterized. We propose that it could be formed by isomerization and hydroxylation. Previous research showed the physically intertwined supercluster genes for the biosynthesis of both pseurotin A and fumagillin. The gene fumR regulates the production of pseurotin A and fumagillin. It was intriguing that the presence of genes in the cluster which were similar to fumagillin targets conferred the strain resistance to fumagillin [60]. However, fumagillin was not isolated from extract of MF071, may be due to low yield.

Several putative prenyltransferases were identified in MF071 for the incorporation of one prenyl moiety in the biosynthesis of fumitremorgins and fumigaclavines, such as Ftm4, Ftm8, Fga3, and Fga8. Fga8 catalysed a “reverse” prenylation of fumigaclavine A with the 2-(1,1-dimethylallyl) moiety connected to the indole system at the 2-position (Figure 6B). The whole genome sequence analysis of MF071 revealed two additional prenyltransferases. Gene deletion experiments or heterologous expression have revealed the function of most “reverse” prenyltransferase genes, such as lxc from Lyngbya majuscule, notF from Aspergillus sp., anaPT from Neosartorya fischeri, brePT from Aspergillus versicolor, and cdpC2PT/cdpNPT from A. nidulans [61-65]. However, the mechanism of the enzymatic catalysis of these “reverse” prenyltransferases has not been fully revealed. The amino acid sequence alignment of Fga8 with the above-mentioned prenyltransferases gave a relatively low similarity value (20%-30%), indicating Fga8 as a potential new prenyltransferase. A possible mechanism of the tert-prenylation at position C2 by Fga 8 was that the prenyltransferase Fga8 initially alkylates the nitrogen atom of the indole. The resulting N-(3,3-dimethylallyl)indole then undergoes an aza-Claisen rearrangement to give the rearranged 3-(1,1-dimethylallyl)indole, followed by a [1,5]-alkyl shift and aromatisation to give the corresponding 2-substituted indole (Figure 7).

Figure 7. Proposed mechanism for the tert-prenylation of indoles at the C-2 position.

It would provide more information to test the substrate specificity of these prenyltransferases as prenylations or tert-prenylations of indole could occur at positions N1, C2, C3, C4, C5, C6,and C7 (Figure S21). Interestingly, the brevianamide F could be catalysed by both FtmB (“regular” prenyltransferase) and NotF to produce tryprostatin B and deoxybrevianamide E, respectively [42,62]. CdpC3PT from A. nidulans has been reported to catalyse the formation of N1-regularly, C2-, and C3- reverse-prenylated derivatives [65]. Further protein structure research could be of importance to confirm this prenylation mechanism.

The current study reports 30 compounds and BGCs of fumitremorgins, pseurotins, fumigaclavines and helvolinic acid, whereas the prediction of MF071 metabolic potential gave large numbers of BGCs. Preliminary blast analysis showed the presence of BGCs for pyripyropene A, neosartoricin B, gliotoxin, trypacidin, xanthocillin, fumisoquin, ferricrocin, 1,8-dihydroxynaphthalene, and many others. However, biosynthetic genes are often silent or transcribed at very low levels under certain conditions, which makes the detection difficult. As the condition used for fermentation is quite different with the native environment (high salinity, oligotrophy, microbial competition, temperature variation, et al.), the chemical profile could be different with that of extract from rice medium fermentation. Approaches learning from nature for activation of these silent BGCs such as OSMAC, microbial co-culture or heterologous expression of unknown clusters could be carried out to further expand the structure classes.

In conclusion, we isolated 30 compounds from Aspergillus fumigatus MF071, including two new compounds 1 and 2. The NMR data of two compounds, monomethylsulochrin-4-sulphate (4) and pseurotin H (10), are also reported here for the first time. Compounds 18 and 19 exhibited strong activities against S. aureus and E. coli. BGCs of fumitremorgins, pseurotins, fumigaclavines and helvolinic acid and biosynthetic pathways were proposed. Mechanism for the tert-prenylation of indoles by prenyltransferase was also discussed.”

  1. In the same way, the discussion should include a conclusion paragraph, where the reader can find a summary of the results that you want to highlight and the main conclusions obtained.

We have provided a conclusion paragraph to summarize the main findings of the current research as following:

“In conclusion, we isolated 30 compounds from Aspergillus fumigatus MF071, including two new compounds 1 and 2. The NMR data of two compounds, monomethylsulochrin-4-sulphate (4) and pseurotin H (10), are also reported here for the first time. Compounds 18 and 19 exhibited strong activities against S. aureus and E. coli. BGCs of fumitremorgins, pseurotins, fumigaclavines and helvolinic acid and biosynthetic pathways were proposed. Mechanism for the tert-prenylation of indoles by prenyltransferase was also discussed.” (line 316-line 321)

  1. Abstract: Don´t use abbreviations (for example BGC)

We have replaced all the abbreviations with full names in the abstract.

  1. Abstract: It should be included a conclusion sentence to summarize the most important findings.

We have added one conclusion sentence to the abstract as suggested by the reviewer: “The current study reported two new compounds and biosynthetic gene clusters of fumitremorgins, pseurotins, fumigaclavines and helvolinic acid from Aspergillus fumigatus MF071” (line 27-29).

  1. In the text there is not a correct use of the BGC abbreviation as sometimes it is abbreviated and sometimes it is not (for example page 2, line 59)

We have carefully reviewed the manuscript and revised all biosynthetic gene clusters with BGCs except for its the first appearance in the manuscript.

  1. There is a recent review about fumagillin, which should be included to explain de intertwin between pseurotin and fumagillin gene clusters and to explain its antimicrobial effect (Guruceaga et al., 2020.Fumagillin, a Mycotoxin of Aspergillus fumigatus:Biosynthesis, Biological Activities, Detection, and Applications. Toxins. 12, 7; doi:10.3390/toxins12010007)

We have taken on board reviewer’s suggestion and included the review paper in the manuscript in regards to the regulation for the production of pseurotin and fumagillin. We also included following paragraph to explain the self-resistant mechanism of fumagillin (line 272-277).

“Previous research showed the physically intertwined supercluster contained genes for biosynthesis of both pseurotin A and fumagillin. The gene fumR regulates the production of pseurotin A and fumagillin. It was intriguing that the presence of genes in the cluster which were similar to fumagillin targets conferred the strain resistance to fumagillin [60]. However, fumagillin was not isolated from extract of MF071, may be due to the low yield.”

In addition, following references have been included in the manuscript.

  1. Shen, L.; Zhu, L.; Luo, Q.; Li, X.; Xi, J.; Kong, G.; Song, Y. Fumigaclavine I, a new alkaloid isolated from endophyte Aspergillus terreus. Chin. J. Nat. Med. 2015, 13, 937-941.
  2. Yadav, R.; Rashid, M. M.; Zaidi, N.; Kumar, R.; Singh, H. Secondary metabolites of Metarhizium spp. and Verticillium spp. and their agricultural applications. In Secondary Metabolites of Plant Growth Promoting Rhizomicroorganisms, Springer: 2019; pp 27-58.
  3. Li, X.; Zhang, Q.; Zhang, A.; Gao, J. Metabolites from Aspergillus fumigatus, an endophytic fungus associated with Melia azedarach, and their antifungal, antifeedant, and toxic activities. J. Agric. Food Chem. 2012, 60, (13), 3424-3431.
  4. Sanmanoch, W.; Mongkolthanaruk, W.; Kanokmedhakul, S.; Aimi, T.; Boonlue, S. Helvolic acid, a secondary metabolite produced by Neosartorya spinosa KKU-1NK1 and its biological activities. Chiang Mai J. Sci. 2016, 43, 483-493.
  5. Ganaha, M.; Yoshii, K.; ÅŒtsuki, Y.; Iguchi, M.; Okamoto, Y.; Iseki, K.; Ban, S.; Ishiyama, A.; Hokari, R.; Iwatsuki, M. In vitro antitrypanosomal activity of the secondary metabolites from the mutant strain IU-3 of the insect pathogenic fungus Ophiocordyceps coccidiicola NBRC 100683. Chem. Pharm. Bull. 2016, 64, (7), 988-990.
  6. Sawadsitang, S.; Mongkolthanaruk, W.; Suwannasai, N.; Sodngam, S. Antimalarial and cytotoxic constituents of Xylaria cf. cubensis PK108. Nat. Prod. Res. 2015, 29, (21), 2033-2036.
  7. Xu, X.; Han, J.; Wang, Y.; Lin, R.; Yang, H.; Li, J.; Wei, S.; Polyak, S. W.; Song, F. Two new spiro-heterocyclic γ-lactams from a marine-derived Aspergillus fumigatus strain CUGBMF170049. Mar. Drugs 2019, 17, (5), 289.
  8. Wenke, J.; Anke, H.; Sterner, O. Pseurotin A and 8-O-demethylpseurotin A from Aspergillus fumigatus and their inhibitory activities on chitin synthase. Biosci. Biotechnol. Biochem. 1993, 57, (6), 961-964.
  9. Guruceaga, X.; Perez-Cuesta, U.; Abad-Diaz de Cerio, A.; Gonzalez, O.; Alonso, R. M.; Hernando, F. L.; Ramirez-Garcia, A.; Rementeria, A. Fumagillin, a mycotoxin of Aspergillus fumigatus: biosynthesis, biological activities, detection, and applications. Toxins 2020, 12, (1), 7.

Reviewer 2 Report

The manuscript presents a lot of new and interesting data on biomolecules obtained from a marine fungal isolate. It is well written and researched, the results are clearly presented, and the discussion is adequate. A few minor edits and additions could improve the manuscript:

  • English language should be checked.

A few examples: line 2: ….chemical exploration of the marine…

Line 43:….infectious pathogens with antimicrobial resistance… (a disease can does not develop resistance, only a pathogen can)

Line 53-54: The genome inspired chemical…(not very clear)

Line 113: (calculated for…)

Line 226: results from Pinheiro et al… (remove first and middle name)

  • The isolate Aspergillus fumigatus is called a marine organism by the authors. The culture used in the experiment was isolated from a marine environment, however A. fumigatus is a common soil fungus found in different environments worldwide. It should be discussed if there are known genetic, morphological , physiological or biochemical differences of marine and terrestrial isolates of A. fumigatus.
  • The authors mention morphological analysis but show only a photo of a culture and report the presence of conidiophores. At least the presence and morphology of conidia should be mentioned.
  • Figure 1: The inclusion of other A. fumigatus strains (in addition to EF669985), esp. from marine habitats would improve the phylogenetic analysis.
  • The chemical profiles were determined for extracts obtained from rice medium fermentation. In order to better understand the chemical profile in the native environment and its role for the fungus, nutritional conditions at the native environment of the isolate should be discussed.
  • Table 2. Title should include: …data of compounds 4 and 6.
  • Line 227: the names of the bacterial strains used by Pinheiro et al. should be listed and the differing results discussed in more detail.

Author Response

Reviewer 2:

The manuscript presents a lot of new and interesting data on biomolecules obtained from a marine fungal isolate. It is well written and researched, the results are clearly presented, and the discussion is adequate. A few minor edits and additions could improve the manuscript:

English language should be checked.

  1. A few examples: line 2: ….chemical exploration of the marine…

We have revised the title as “Genome-inspired chemical exploration of marine fungus Aspergillus fumigatus MF071” according to the reviewer’s suggestion.

  1. Line 43:….infectious pathogens with antimicrobial resistance… (a disease can does not develop resistance, only a pathogen can)

We have revised the sentence from

“The increasing rediscovery rate of known compounds through high-throughput screening (HTS) has led to a decline in natural product research, whereas both hospital and community-associated infectious diseases with antimicrobial resistance (AMR) are spreading rapidly [10]”

to

“The increasing rediscovery rate of known compounds through high-throughput screening (HTS) has led to a decline in natural product research, whereas both hospital and community-associated infectious pathogens with antimicrobial resistance (AMR) are spreading rapidly [10].”

  1. Line 53-54: The genome inspired chemical…(not very clear)

We have revised the sentence from

“The genome inspired chemical and biological investigation of the strain led to the isolation of 30 compounds including the two new compounds 19S,20-epoxy-18-oxotryprostatin A (1) and 20-hydroxy-18-oxotryprostatin A (2).”

to

“The genome-inspired chemical constituents study and biological activities investigation of extract of MF071 led to the isolation of 30 compounds, including two new compounds 19S,20-epoxy-18-oxotryprostatin A (1) and 20-hydroxy-18-oxotryprostatin A (2), along with seven active compounds.”

  1. Line 113: (calculated for…)

We have revised “calcd. for” to “calculated for” throughout the manuscript.

  1. Line 226: results from Pinheiro et al… (remove first and middle name)

We have removed first and middle name accordingly.

  1. The isolate Aspergillus fumigatus is called a marine organism by the authors. The culture used in the experiment was isolated from a marine environment, however A. fumigatus is a common soil fungus found in different environments worldwide. It should be discussed if there are known genetic, morphological, physiological or biochemical differences of marine and terrestrial isolates of A. fumigatus.

Authors acknowledge reviewer’s insightful comments on the origin of Aspergillus fumigatus. It is true that A. fumigatus is widespread and easily adapted to diverse environment, and occupied diverse habitats because of its easily dispersed conidia. Literature review suggests that Aspergillus fumigatus is also frequently found in marine environment [1]. Diverse secondary metabolites have been isolated from both terrestrial and marine-derived A. fumigatus [2-6]. This observation is very much supported by our current study.

  1. The authors mention morphological analysis but show only a photo of a culture and report the presence of conidiophores. At least the presence and morphology of conidia should be mentioned.

Taking on board reviewer’s comment, we have revised the original statement from

“After incubation at 28 °C for ten days, strain MF071 formed colonies on a PDA plate with specialized hyphal structures called conidiophores (Figure 1a)”

to

“After incubation at 28 °C for ten days, strain MF071 formed colonies on a PDA plate with characteristic hyphal structures (Figure 1a)”

  1. Figure 1: The inclusion of other A. fumigatus strains (in addition to EF669985), esp. from marine habitats would improve the phylogenetic analysis.

We have revised Figure 1 by including two other A. fumigatus strains, Aspergillus fumigatus IHEM 18963 (KP131566) and Aspergillus fumigatus FSU6405 (GQ221107) and regenerated the phylogenetic tree.

  1. The chemical profiles were determined for extracts obtained from rice medium fermentation. In order to better understand the chemical profile in the native environment and its role for the fungus, nutritional conditions at the native environment of the isolate should be discussed.

We thank reviewer’s suggestions and revised the discussion section accordingly.

“As the condition used for fermentation is quite different with the native environment (high salinity, oligotrophy, microbial competition, temperature variation, et al.), the chemical profile could be different with that of extract from rice medium fermentation. Approaches learning from nature for activation of these silent BGCs such as OSMAC, microbial co-culture or heterologous expression of unknown clusters could be carried out to further expand the structure classes.” (line 310-line 315)

We will endeavor to investigate the chemical profile of the extract cultured in the native environment as part of future work.

  1. Table 2. Title should include: …data of compounds 4 and 6.

We have revised the titles of table 1, table 2, and table 3 accordingly:

Table 1 1H (800 MHz) and 13C (200 MHz) NMR data of compounds 1 and 2

Table 2 1H (800 MHz) and 13C (200 MHz) NMR data of compounds 4 and 6

Table 3. 1H (800 MHz) and 13C (200 MHz) NMR data of compound 10

  1. Line 227: the names of the bacterial strains used by Pinheiro et al. should be listed and the differing results discussed in more detail.

We have included the names and ATCC ID of bacterial strains in “4.6 Bioassays” section. A discussion on the results have also been included as following (line 266-270):

“However, the compounds showed no antibacterial activity in our screening at concentrations up to 100 μg/mL, which did not agree with the results of antibacterial activity against E. coli (ATCC 25922), P. aeruginosa (ATCC 27853), S. aureus (ATCC 25923) from Pinheiro et al [57]. It is likely that different bacterial strains contributed to the different results as pseurotin A was also reported to have no activity against S. aureus (ATCC 6538) and S. aureus [58,59].”

References

  1. Lee, S.; Park, M. S.; Lim, Y. W. Diversity of marine-derived Aspergillus from tidal mudflats and sea sand in Korea. Mycobiology 2016, 44, (4), 237-247.
  2. Lee, Y. M.; Kim, M. J.; Li, H.; Zhang, P.; Bao, B.; Lee, K. J.; Jung, J. H. Marine-derived Aspergillus species as a source of bioactive secondary metabolites. Mar. Biotechnol. 2013, 15, (5), 499-519.
  3. Afiyatullov, S. S.; Zhuravleva, O. I.; Antonov, A. S.; Kalinovsky, A. I.; Pivkin, M. V.; Menchinskaya, E. S.; Aminin, D. L. New metabolites from the marine-derived fungus Aspergillus fumigatus. Nat. Prod. Commun. 2012, 7, (4), 1934578X1200700421.
  4. Han, X.; Xu, X.; Cui, C.; Gu, Q. Alkaloidal compounds produced by a marine-derived fungus, Aspergillus fumigatus H1-04, and their antitumor activities. Chinese Journal of Medicinal Chemistry 2007, 17, (4), 232.
  5. Li, Y.-X.; Himaya, S.; Dewapriya, P.; Kim, H. J.; Kim, S.-K. Anti-proliferative effects of isosclerone isolated from marine fungus Aspergillus fumigatus in MCF-7 human breast cancer cells. Process Biochem. 2014, 49, (12), 2292-2298.
  6. Zhao, W. Y.; Zhu, T. J.; Fan, G. T.; Liu, H. B.; Fang, Y. C.; Gu, Q. Q.; Zhu, W. M. Three new dioxopiperazine metabolites from a marine-derived fungus Aspergillus fumigatus Fres. Nat. Prod. Res. 2010, 24, (10), 953-957.

In addition, following references have been included in the manuscript.

  1. Shen, L.; Zhu, L.; Luo, Q.; Li, X.; Xi, J.; Kong, G.; Song, Y. Fumigaclavine I, a new alkaloid isolated from endophyte Aspergillus terreus. Chin. J. Nat. Med. 2015, 13, 937-941.
  2. Yadav, R.; Rashid, M. M.; Zaidi, N.; Kumar, R.; Singh, H. Secondary metabolites of Metarhizium spp. and Verticillium spp. and their agricultural applications. In Secondary Metabolites of Plant Growth Promoting Rhizomicroorganisms, Springer: 2019; pp 27-58.
  3. Li, X.; Zhang, Q.; Zhang, A.; Gao, J. Metabolites from Aspergillus fumigatus, an endophytic fungus associated with Melia azedarach, and their antifungal, antifeedant, and toxic activities. J. Agric. Food Chem. 2012, 60, (13), 3424-3431.
  4. Sanmanoch, W.; Mongkolthanaruk, W.; Kanokmedhakul, S.; Aimi, T.; Boonlue, S. Helvolic acid, a secondary metabolite produced by Neosartorya spinosa KKU-1NK1 and its biological activities. Chiang Mai J. Sci. 2016, 43, 483-493.
  5. Ganaha, M.; Yoshii, K.; ÅŒtsuki, Y.; Iguchi, M.; Okamoto, Y.; Iseki, K.; Ban, S.; Ishiyama, A.; Hokari, R.; Iwatsuki, M. In vitro antitrypanosomal activity of the secondary metabolites from the mutant strain IU-3 of the insect pathogenic fungus Ophiocordyceps coccidiicola NBRC 100683. Chem. Pharm. Bull. 2016, 64, (7), 988-990.
  6. Sawadsitang, S.; Mongkolthanaruk, W.; Suwannasai, N.; Sodngam, S. Antimalarial and cytotoxic constituents of Xylaria cf. cubensis PK108. Nat. Prod. Res. 2015, 29, (21), 2033-2036.
  7. Xu, X.; Han, J.; Wang, Y.; Lin, R.; Yang, H.; Li, J.; Wei, S.; Polyak, S. W.; Song, F. Two new spiro-heterocyclic γ-lactams from a marine-derived Aspergillus fumigatus strain CUGBMF170049. Mar. Drugs 2019, 17, (5), 289.
  8. Wenke, J.; Anke, H.; Sterner, O. Pseurotin A and 8-O-demethylpseurotin A from Aspergillus fumigatus and their inhibitory activities on chitin synthase. Biosci. Biotechnol. Biochem. 1993, 57, (6), 961-964.
  9. Guruceaga, X.; Perez-Cuesta, U.; Abad-Diaz de Cerio, A.; Gonzalez, O.; Alonso, R. M.; Hernando, F. L.; Ramirez-Garcia, A.; Rementeria, A. Fumagillin, a mycotoxin of Aspergillus fumigatus: biosynthesis, biological activities, detection, and applications. Toxins 2020, 12, (1), 7.

Round 2

Reviewer 1 Report

My main concerns are still two:

  • It cannot be discarded that the strain is not an environmental contamination.
  • It is interesting that some new molecules have been found and described, but it seems that the strain is not different from the environmental strains. So, the fact that it is a marine strain is not relevant for the study. It should be explained.

Minor comment:

The introduced sentence "However, some of them were also reported from taxonomically close species. such as..." has a dot that should be removed.

Author Response

My main concerns are still two:

It cannot be discarded that the strain is not an environmental contamination.

It is interesting that some new molecules have been found and described, but it seems that the strain is not different from the environmental strains. So, the fact that it is a marine strain is not relevant for the study. It should be explained.

We acknowledge reviewer’s insightful comments on the origin of Aspergillus fumigatus. When we carried out the isolation experiment, we used “sterile artificial seawater” as control to avoid the contamination during our isolation process, and we didn’t see any colony from the control medium plate. We are confident there are no contamination during the isolation process. We added and highlighted the control we used in “4.2. Microbial Strain Culture and Identification”:

“Specifically, 1.0 g of sediment sample was added into 50 mL sterile centrifuge tube and suspended in 9 mL sterile artificial seawater (3.8% sea salt) under aseptic operation. An aliquot of 200 µL diluted suspension (1/10) was spread plated on the separation medium (1.0% peptone, 4.0% glucose, 1.5% agar, pH 6.0) supplemented with 0.5 mg/mL chloramphenicol and streptomycin, and 200 µL sterile artificial seawater was also spread plated on another plate as control.” (line 323-324)

However, it is true that A. fumigatus is widespread and easily adapted to diverse environment, and occupied diverse habitats because of its easily dispersed conidia. Literature review suggests that A. fumigatus is also frequently found in marine environment [1-6]. We agree that the contamination from the sampling process can not be excluded. Therefore, one paragraph of postscript was included after the discussion:

Postscript

A reviewer has commented that the strain of A. fumigatus used might be the result of environmental contamination. We agree that this is a possibility as A. fumigatus occurs widely in the environment including in soil and plant matter. It is possible that the A. fumigatus that was obtained from a marine sediment collected at a depth of 60 m from the Bohai Sea, China, has a terrestrial origin.”

Minor comment:

  1. The introduced sentence "However, some of them were also reported from taxonomically close species. such as..." has a dot that should be removed.

The dot in this sentence has been removed.

References

  1. Lee, S.; Park, M. S.; Lim, Y. W. Diversity of marine-derived Aspergillus from tidal mudflats and sea sand in Korea. Mycobiology 2016, 44, (4), 237-247.
  2. Lee, Y. M.; Kim, M. J.; Li, H.; Zhang, P.; Bao, B.; Lee, K. J.; Jung, J. H. Marine-derived Aspergillus species as a source of bioactive secondary metabolites. Marine biotechnology 2013, 15, (5), 499-519.
  3. Afiyatullov, S. S.; Zhuravleva, O. I.; Antonov, A. S.; Kalinovsky, A. I.; Pivkin, M. V.; Menchinskaya, E. S.; Aminin, D. L. New metabolites from the marine-derived fungus Aspergillus fumigatus. Natural product communications 2012, 7, (4), 1934578X1200700421.
  4. Han, X.-x.; Xu, X.-y.; Cui, C.-b.; Gu, Q.-q. Alkaloidal compounds produced by a marine-derived fungus, Aspergillus fumigatus H1-04, and their antitumor activities. Chinese Journal of Medicinal Chemistry 2007, 17, (4), 232.
  5. Li, Y.-X.; Himaya, S.; Dewapriya, P.; Kim, H. J.; Kim, S.-K. Anti-proliferative effects of isosclerone isolated from marine fungus Aspergillus fumigatus in MCF-7 human breast cancer cells. Process Biochemistry 2014, 49, (12), 2292-2298.
  6. Zhao, W. Y.; Zhu, T. J.; Fan, G. T.; Liu, H. B.; Fang, Y. C.; Gu, Q. Q.; Zhu, W. M. Three new dioxopiperazine metabolites from a marine-derived fungus Aspergillus fumigatus Fres. Nat. Prod. Res. 2010, 24, (10), 953-957.